# Aerobic exercise and brain structure among military service members and Veterans with varying histories of mild traumatic brain injury: A LIMBIC-CENC exploratory investigation

Samuel R. Walton[1,2]*, John J. Fraser[3,4], Jessie R. Oldham[1], Mark L. Ettenhofer[5,6,7], Patrick Armistead-Jehle[8], Hannah M. Lindsey[9,10], Naomi J. Goodrich-Hunsaker[9,10], Emily L. Dennis[9], Elisabeth A. Wilde[9,10], Sarah M. Jurick[11,12], Michael J. Hall[1], Randel L. Swanson[13,14], Andrew J. MacGregor[11], David F. Tate[9], David X. Cifu[1,2], William C. Walker[1,2]

1 Department of Physical Medicine and Rehabilitation, Virginia Commonwealth University School of Medicine, Richmond, Virginia, United States of America, 2 Richmond Veterans Affairs Medical Center, Central Virginia Virginia Health Care System, Richmond, Virginia, United States of America, 3 Department of Physical Medicine and Rehabilitation, Uniformed Services University of the Health Sciences School of Medicine, Bethesda, Maryland, United States of America, 4 University of Kentucky Sports Medicine Research Institute, Lexington, Kentucky, United States of America, 5 Traumatic Brain Injury Center of Excellence, Naval Medical Center San Diego, San Diego, California, United States of America, 6 University of California, San Diego, California, United States of America, 7 General Dynamics Information Technology, San Diego, California, United States of America, 8 Munson Army Health Center, Fort Leavenworth, Kansas, United States of America, 9 Department of Neurology, University of Utah School of Medicine, Salt Lake City, Utah, United States of America, 10 George E. Wahlen Virginia Medical Center, Salt Lake City, Utah, United States of America, 11 Naval Health Research Center, San Diego, California, United States of America, 12 Leidos, Inc., Reston, Virginia, United States of America, 13 Department of Physical Medicine and Rehabilitation, University of Pennsylvania Perelman School of Medicine, Philadelphia, Pennsylvania, United States of America, 14 Center for Neurotrauma, Neurodegeneration and Restoration, Corporal Michael J. Crescenz Virginia Medical Center, Philadelphia, Pennsylvania, United States of America

* samuel.walton@vcuhealth.org (SRW)

## Abstract

### Objectives

To explore associations of recent moderate-to-vigorous aerobic exercise (MVAE) participation and lifetime mild traumatic brain injury (mTBI) history with measures of brain gray matter volumes among military service members and Veterans (SMVs).

### Methods

Participants ($n = 1,340$; aged $41.3 \pm 10.3$ years; 13% female) were SMV's who participated in the Long-term Impact of Military-relevant Brain Injury Consortium-Chronic Effects of Neurotrauma Consortium Prospective Longitudinal Study (LIMBIC-CENC PLS). MVAE participation was self-reported via the Behavioral Risk Factor Surveillance System and categorized according to current MVAE recommendations (*Inactive, Insufficiently Active, Active, and Highly Active*). Lifetime mTBI history was queried via validated structured

**Data availability statement:** All study data are available free of charge via FITBIR (https://fitbir.nih.gov/) pending successful application for access according to FITBIR procedures (https://fitbir.nih.gov/content/access-data). Data processing and analysis procedures and/or code may be made available upon reasonable request to the corresponding author.

**Funding:** MLE, SMJ, JJF, AJM, PAJ are military service member employees of the United States government. This work was prepared as part of their official duties. Title 17 U.S.C. 105 provides that copyright protection under this title is not available for any work of the United States Government. This work was supported by the Assistant Secretary of Defense for Health Affairs endorsed by the Department of Defense, through the Psychological Health/Traumatic Brain Injury Research Program Long-Term Impact of Military-Relevant Brain Injury Consortium (LIMBIC) Award/W81XWH-18-PH/TBIRP-LIMBIC under Awards No. W81XWH1920067 and W81XWH-13-2-0095, and by the U.S. Department of Veterans Affairs Awards No. I01 CX002097, I01 CX002096, I01 HX003155, I01 RX003444, I01 RX003443, I01 RX003442, I01 CX001135, I01 CX001246, I01 RX001774, I01 RX 001135, I01 RX 002076, I01 RX 001880, I01 RX 002172, I01 RX 002173, I01 RX 002171, I01 RX 002174, and I01 RX 002170. The U.S. Army Medical Research Acquisition Activity, 839 Chandler Street, Fort Detrick MD 21702-5014 is the awarding and administering acquisition office. The study protocol was approved by the Virginia Commonwealth University Institutional Review Board in compliance with all applicable federal regulations governing the protection of human subjects, protocol number HM20002321. The views, opinions, interpretations, conclusions and recommendations expressed are those of the authors and do not necessarily reflect the official policy of the Department of the Navy, the Department of the Army, Department of Defense, Department of Veterans Affairs or the U.S. Government. The funders had no role in study design, data collection and analysis, decision to publish, or preparation of the manuscript.

**Competing interests:** I have read the journal's policy and the authors of this manuscript have the following competing interests: SRW declares separate funding support from

interview and categorized as *0 mTBI*, *1-2 mTBIs*, *3+ mTBIs*. Structural MRI (T1- and T2-weighted images) were used to measure gray matter volumetrics: ventricle-to-brain ratio (VBR); bilateral volumes of the frontal, parietal, temporal, occipital, cingulate, hippocampus, amygdala, and thalamus regions. Multivariable linear regression models were fit to test associations of MVAE participation, mTBI history, and their interaction on each of the volumetric outcomes while controlling for age, sex, education attainment, and PTSD symptoms. Effects were considered statistically significant if the corresponding unstandardized beta (B) and 95% CI did not include 0.

## Results

Regarding main effects, participants in the *Inactive* MVAE group had significantly larger VBR values (worse outcome) than those in the *Insufficiently Active* group (B[95%CI] = -0.137[-0.260, -0.014]). Interaction effects showed participants with no lifetime mTBIs in the *Highly Active* group had larger VBR values (worse outcome) when compared to those in the *Inactive and Insufficiently Active* groups. SMVs with 3+ lifetime mTBIs who were *Highly Active* also had smaller VBR values (better outcome) when compared to *Highly Active* SMVs with fewer lifetime mTBIs. There were no other statistically significant differences for MVAE participation, mTBI history, or their interactions.

## Conclusions

History of one or more lifetime mTBIs was not associated with measures of brain gray matter volumes, suggesting that declines in structural brain health are not expected for the most SMVs with mTBI(s). Although MVAE may benefit brain health, a positive association between self-reported MVAE participation and gray matter volumes was not observed.

## Introduction

Over 400,000 mild traumatic brain injuries (mTBIs) have been documented in active-duty military service members since the year 2000 [1]. Recent studies have reported that subsets of individuals with a history of one or more mTBI events experience advanced brain aging [2], diminished cortices [3–6], subcortical gray matter atrophy [7,8], or higher prevalence of neurodegenerative diseases [9–13]. These reports suggest that maladaptive changes to brain gray matter could result from mTBI exposure, which may subsequently result in progressive functional declines; however, this causal association is not well-supported in the literature [14,15]. Brain gray matter volumetrics have been identified as a viable surrogate measure of brain reserve, which is operationally defined as the availability of physical neural resources that mitigate the effects of neural decline caused by aging or age-related disease [16]. Therefore, exposure to one or more mTBI events may have deleterious effects on brain reserve, and if left unaddressed, these changes in gray matter may contribute to accelerated or pathological declines in brain health.

Aerobic exercise is a readily-available and self-activating potential intervention for promoting brain health among individuals with exposure to mTBI [17,18]. A recent systematic review and meta-analysis of exercise interventions for athletes with sport-related concussion (a type of mTBI) [19] noted the safety and efficacy of aerobic exercise prescriptions for facilitating adaptive recovery [20]. However, most research evaluating exercise as an intervention after mTBI has focused on adolescent and young adult athletes in the acute or subacute phases of

the Departments of Defense and Veterans affairs for projects related to the LIMBIC-CENC study; honorarium and conference travel support from the National Athletic Trainers' Association and the University of California – Los Angeles; and serves in unpaid leadership roles with the World Federation of Athletic Training and Therapy (WFATT) and the Concussion in Sport Group (CISG). JJF reports grants from Congressionally Directed Medical Research Programs and the Office of Naval Research, outside of the submitted work; and has a patent pending for an Adaptive and Variable Stiffness Ankle Brace, U.S. Provisional Patent Application No. 63254,474. JRO receives support from the Thrasher Research Fund and National Institute of Arthritis and Musculoskeletal Skin Diseases (8K12AR084233-03). EAW reports grants from Congressionally Directed Medical Research Programs, US Department of Veterans Affairs, and National Institutes of Health, outside of the submitted work. RLS was supported, in part, by the US Department of Veterans Affairs Rehabilitation Research and Development Service under award number IK2 RX003651, for an independent project related to the LIMBIC-CENC study. DFT, DXC, and WCW report separate funding support from the DoD & VA for projects related to the LIMBIC-CENC study. The remaining authors have no disclosures of potential conflicts of interest to report. This does not alter our adherence to PLOS ONE policies on sharing data and materials.

mTBI recovery (i.e., a few days to a few weeks after injury). Current standard of care recommendations for patients with mTBI outside that population do not routinely include aerobic exercise prescription, nor do they address exercise beyond the acute phase of recovery [21]. In healthy individuals, aerobic exercise has been shown to positively influence brain morphology from youth to older adults [22,23]. Specifically, participation in moderate-to-vigorous aerobic exercise (MVAE) can preserve cortical [24–26] and subcortical gray matter in aging populations [26–29], and may prevent the onset of progressive neurodegenerative changes and their affiliated symptoms [28,30–33]. Further, among both healthy older adults and those with neurodegenerative disease, MVAE interventions over several months preserved brain gray matter and improve cognitive test performance, potentially through improved cardiovascular fitness [24,27,28,33]. Clinical intervention using MVAE prescriptions in populations with relatively high exposure to mTBI, such as military service members and Veterans (SMVs), may be able to counter the potential deleterious effects of these injuries among many of the affected brain regions (e.g., frontal cortex, hippocampus). Early evidence from investigations with SMVs and former American football players have reported positive associations between MVAE participation and subjective measures of cognitive and mood-related functioning, despite history of exposure to one or more occupation-related mTBIs [34–36]. Still, the converging effects of mTBI history and aerobic physical activity are understudied. Additional research in this area is necessary to inform evidence-based application of MVAE as a modifiable behavior and practical therapy to improve brain health.

The purpose of this retrospective cohort study was to test the unique and interactive effects of mTBI history and MVAE participation on surrogate measures of brain reserve among the brain as a whole and parsed by cortical and subcortical gray matter regions. We hypothesized that: 1) a greater number of lifetime mTBIs would be associated with diminished brain reserve (e.g., smaller gray matter volumes); and 2) higher weekly MVAE engagement would be associated with preserved brain reserve (e.g., larger gray matter volumes). We also anticipated that this association would be present across all levels of lifetime mTBI history (i.e., those reporting 0, 1-2, and 3 or more lifetime mTBIs).

## Materials & methods

### Participants

This study was a secondary analysis of data from a cohort of individuals with and without mTBI enrolled in the Long-term Impact of Military-relevant Brain Injury Consortium-Chronic Effects of Neurotrauma Consortium Prospective Longitudinal Study (LIMBIC-CENC PLS) between January 6th, 2015 and September 30th, 2022. The Strengthening the Reporting of Observational Studies in Epidemiology (STROBE) was used to guide this report [37]. The specific details of the LIMBIC-CENC PLS data collection methods have been previously reported [38,39]. Briefly, the LIMBIC-CENC PLS captures multifaceted information about key brain health factors of combat-exposed SMVs, including injury history (e.g., number and mechanisms of lifetime mTBIs), validated self-report questionnaires of health and wellbeing (e.g., depressive symptoms, social participation) and participation in health-related behaviors (e.g., MVAE, alcohol consumption), objectively measured cognitive functioning (e.g., neuropsychological testing performance), physiologic functioning (e.g., posturography, vision testing), and biomarkers (e.g., multimodal MRI and blood-based measures). Inclusion criteria for participation in the ongoing LIMBIC-CENC PLS are: 1) at least 18 years of age; 2) military deployment to a combat zone; and 3) combat exposure as defined by the Deployment Risk and Resiliency Inventory Section D (DRRI-2-D) score >1 on any item [40]. Exclusion criteria are: 1) any history of moderate, severe, or penetrating TBI; or 2) history of major neurologic

disorder or psychiatric disorder causing a significant reduction in independent living status (e.g., schizophrenia). Note that TBI-negative control participants were included, as were those with common mental health or medical conditions (e.g., PTSD, depression, hypertension, chronic pain). The LIMBIC-CENC PLS was approved by the Institutional Review Boards at all participating research sites and each participant provided their written informed consent prior to engaging in any research activities.

In the present study, we included all participants with data available for each of the primary predictors of interest (lifetime mTBI history, participation in MVAE) and pertinent MRI images needed for evaluation of brain structure (i.e., T1-weighted images or T1- and T2-weighted images). Of the $n = 2,263$ participants who completed the parent LIMBIC-CENC PLS baseline study assessment between 2015-2022, none were missing lifetime mTBI history information, $n = 914$ (40.1%) did not have MRI data available (e.g., did not undergo MRI assessment), and $n = 12$ (0.5%) were missing MVAE participation information, but were already included in the subset who did not have MRI data.

Given the number of participants who lacked MRI data, we performed analyses to determine whether those with versus without MRI data differed on multiple study-relevant variables including: age; sex; mTBI group; MVAE participation; highest level of education completed; active-duty vs. Veteran status; exposure to a blast-related mTBI mechanism; and time since most recent mTBI. There was a statistically significant, but minimal difference in age between these groups (mean age difference = 1 year; $p = 0.02$; Cohen's $d$ [95% CI] = 0.10 [.02 -.18]). Additionally, among participants with at least one lifetime mTBI, the proportion of participants with one or more blast-related mTBI in the group with MRI data was 5% lower than among those without available MRI data (43.7% vs. 48.6%, respectively; $X^2_{(1)} = 4.378$, $p = 0.04$). More detailed results from these analyses are provided in S1 Supplemental Methods.

## Measurement of lifetime mTBI exposure

Lifetime mTBI history was captured and categorized through a validated structural interview and rigorous data integrity review process. Using the VCU-rCDI [41], each participant reported potential concussive events (PCEs) that were then categorized through a diagnostic interview and subsequent algorithm-based classification as *no mTBI*, *mTBI with posttraumatic amnesia*, or *mTBI without posttraumatic amnesia*. Every preliminary algorithm classification was reviewed by the site PI, compared to all interview elements including unstructured free-text data and available medical records, and their final determinations were further vetted by a central study monitor and, if needed, by consensus of an expert committee [41]. The referent definition of mTBI was based on the VA/DoD common definition of mTBI, which aligns with the American College of Rehabilitation Medicine's definition of mTBI [42–44]. In the present study, similar to related work with an overlapping sample [14], the total number of lifetime mTBIs was operationalized into three separate groups for analyses: those reporting no lifetime mTBIs (*0 mTBI*); those reporting one or two lifetime mTBIs (*1-2 mTBI*); those reporting three or more lifetime mTBIs (*3+ mTBIs*).

## Measurement of MVAE participation

Participants reported whether they participated in regular physical activity or exercise during the last month on the Centers for Disease Control and Prevention's Behavioral Risk Factor Surveillance System (BRFSS) questionnaire [45]. If they indicated that they did participate in leisure-time activities or structured exercise, they then selected the primary type of activity (e.g., running, swimming) along with the frequency (weekly) and duration (in minutes) of these typical activities [46]. The User Guidelines for the BRFSS provide metabolic equivalent

(MET) values for each activity to categorize them as non-aerobic, light aerobic, moderate aerobic, or vigorous aerobic intensities [47]. For each participant's primary activity, we calculated the total amount of time spent per week engaged at that aerobic intensity, and we subsequently categorized participants into MVAE groups according to the following guidelines from the Department of Health and Human Services (DHHS) [48]: those that did not engage in any aerobic physical activity (*Inactive*); those that did engage in aerobic physical activity but did not meet DHHS recommendations (*Insufficiently Active*); those that met the minimal DHHS recommendation of 150 minutes per week of moderate intensity aerobic activity, 75 minutes of vigorous aerobic activity, or equivalent combination thereof (*Active*); and, those that exceeded the minimal DHHS guidelines by at least twice the recommended volume of aerobic physical activity per week (*Highly Active*). We acknowledge that inclusion of additional activities may have affected the ultimate allocation into MVAE groups; however, additional activities were not reported by a large portion of the sample with available MRI data ($n$ = 692/1,349; 51.3%), and therefore only primary physical activity participation was considered in the present study.

For planned sensitivity analyses, we also calculated the total number of MET-minutes per week (MMpW) by multiplying primary activity MET values by the total number of minutes engaged in that particular activity as a way of creating a continuous variable for aerobic activity engagement [48]. This operationalization also allows for consideration of the health benefits from light intensity aerobic activity on health outcomes alongside the noted value of MVAE, even if it does not meet the aforementioned MVAE guidelines. Total MMpW values were available for all participants in the Highly Active and Active groups; however, there were missing MET values for activities reported by $n$ = 15 (3.5%) of the Insufficiently Active group and $n$ = 323 (70.7%) of the Inactive group, reflecting the inability to crosswalk all participant-reported physical activities with the BRFSS User Guidelines (i.e., some activities reported by participants were unclear or were not accounted for in the BRFSS User Guidelines). Therefore, these sensitivity analyses included only the subset of study participants ($n$ = 1,011; 74.9%) who reported a primary physical activity that had an allocated MET value.

## MR image acquisition, processing, and outcome variables

MRI acquisition: Participants underwent multimodal 3T MRI assessment, including T1- and T2-weighted images, at one of 10 participating clinical research sites. MRI acquisition followed a specific protocol that was overseen by a centralized Neuroimaging Core for compliance and quality assurance. Acquisition parameters for participating sites have been reported previously [49].

MRI data processing: MRI data preprocessing and processing were conducted by HML, NJG, ELD, and EAW utilizing resources and support provided by the Center for High-Performance Computing (CHPC) at the University of Utah. T1- and T2-weighted DICOM files obtained were converted to the NIfTI format using the dcm2niix tool available in MRIcron (https://github.com/rordenlab/dcm2niix). Next, the T1-weighted images underwent standard alignment utilizing the acpcdetect tool from the Automatic Registration Toolbox (available at www.nitrc.org/frs/?group_id=90). For all T1-weighted data, intensity correction was carried out using Advanced Normalization Tools (ANTs) version 2.4.1 N4 Bias Field Correction [50]. Subsequently, the T2-weighted images for each participant were registered to their corresponding T1-weighted images using the ANTs registration method, antsRegistrationSyNQuick.

Once preprocessing was completed, FreeSurfer v7.1.1 was used to automatically parcellate cortical and segment subcortical brain regions from the co-registered T1- and T2-weighted

anatomical images (freely available at http://surfer.nmr.mgh.harvard.edu). Cortical parcella-
tions were identified and labeled within the surface-based processing stream according to the
Desikan-Kiliany atlas definitions of gyri and sulci for each hemisphere.

Structural outcome measures: Structural variables of interest in this study were: ventricle-
to-brain ratio (VBR); bilateral composite cortical volumes of the frontal lobes, parietal lobes,
temporal lobes, occipital lobes, and cingulate cortices; and subcortical structure volumes of
the hippocampus, amygdala, and thalamus. Total ventricle volume was calculated as the sum
of volumes for bilaterally measured lateral ventricles, inferior lateral ventricles, and cho-
roid plexi added to the $3^{rd}$, $4^{th}$, and $5^{th}$ ventricle volumes. VBR was calculated as the ratio of
total ventricular volume to total brain segmentation volume. Total frontal lobe volume was
calculated as the sum of the 10 left and 10 right frontal lobe sub-volumes. Total parietal lobe
volume was calculated as the sum of the six left and six right parietal lobe sub-volumes. Total
temporal lobe volume was calculated as the sum of the nine left and nine right temporal lobe
sub-volumes. Total occipital lobe volume was calculated as the sum of the four left and four
right occipital lobe sub-volumes. Total cingulate cortex volume was calculated as the sum of
the five left and five right cingulate cortex composite volumes. Total hippocampal volume was
calculated as the sum of the 19 left and 19 right hippocampal subfields. Total amygdala vol-
ume was calculated as the sum of the nine left and nine right amygdalar nuclei. Total thalamus
volume was calculated as the sum of the 24 left and 24 right thalamic nuclei. Specific bilateral
cortical sub-volumes, hippocampal subfields, and amygdalar and thalamic nuclei are reported
in the Table in S1 Supplemental Materials.

## Selection of covariates for analyses

Multiple potential confounding variables were selected from demographic and behavioral data
collected during the same study visit. These variables were: age (years), self-reported sex/gen-
der (dichotomous operationalization; "Are you female or male?"), highest level of education
completed (ordinal: completed some or all of high school; completed 1 to 3 years of college
or technical school; completed at least a 4-year collegiate education), hazardous alcohol use
(dichotomous; scores > 3 for males/ >2 for females on the Alcohol Use Disorders Identifica-
tion Test-Concise), and meeting screening criteria for posttraumatic stress symptoms on the
Posttraumatic Stress Syndrome (PTSD) Checklist for the Diagnostic and Statistical Manual,
fifth edition (PCL-5; total score > 33) [51]. Variables that were observed to have statistically
significant associations (test $p$-value < 0.05) with both a predictor of interest (mTBI history
and/or MVAE participation) and one or more of the primary study outcomes (brain volumet-
ric measures) were considered as potential confounders and were subsequently included as
covariates in all analyses. Age, sex, education, and PTSD symptoms were deemed to be poten-
tial confounders by these criteria, but hazardous alcohol use was not observed to be related to
either mTBI history or MVAE participation in this sample.

## Statistical analyses

Among the participants with complete predictor and outcome data ($n$ = 1,349), $n$ = 43 (3.2%)
were observed to have unusable images (e.g., motion artifacts) or incidental findings (e.g.,
white matter lesions) on clinical readings of their structural MRI images and were subse-
quently removed from study analyses. Nine participants were missing data for PTSD
symptoms, but not for any of the other covariates (age, sex, education attainment). Of the
remaining participants ($n$ = 1,340), structural outcomes were based on T1-weighted images
only for $n$ = 41 (3.1%) participants, and the remaining participants ($n$ = 1,299) had both
T1-weighted and T2-weighted images. The difference in the proportion of participants within

mTBI history groups and MVAE groups that had only T1-weighted images were not statistically significant (mTBI: $X^2(2) = 5.849$; $p = 0.05$; MVAE: $X^2(3) = 2.179$; $p = 0.54$), and therefore, we included all participants regardless of the availability of T2-weighted images.

To test our hypotheses that more lifetime mTBIs and less MVAE participation were associated with larger VBR and smaller gray matter region volumes, a multivariable general linear model was fit for each outcome. Prior to performing study analyses, all outcome variables were transformed into *z*-scores based on the sample distribution (sample $M \pm SD. = 0.0 \pm 1.0$). Primary predictors were evaluated using simple contrasts. For mTBI history groups, we compared those with 1-2 lifetime mTBIs and those with 3 + lifetime mTBIs to those with 0 lifetime mTBIs. For MVAE, *Insufficiently Active*, *Active*, and *Highly Active* groups were each compared to those in the *Inactive* group. Participant age, sex, highest level of education, and PTSD symptoms were included as covariates in all models. Unstandardized beta coefficients (*B*) and 95% confidence intervals (CI) were calculated for each predictor in each model, and were considered statistically significant if the 95% CI did not include zero. Next, to test whether the effects of MVAE participation were dependent on lifetime mTBI history, an interaction term was added to the aforementioned models. Overall interaction effects were tested with Wald chi-squared ($X^2$) tests and were considered statistically significant if the interaction term *p*-value < 0.05. When a statistically significant interaction was observed for an outcome, post-hoc pairwise comparisons were made between MVAE groups within each level of lifetime mTBI history to determine whether marginal means and Wald 95% CIs were different between participants in the *Inactive* (referent) group and those in each of the other MVAE groups.

For sensitivity analyses regarding the operationalization of aerobic exercise, we explored aerobic exercise participation as a continuous variable using MMpW, z-scored from data within our sample. We included this derived variable in the aforementioned main effects models in place of the categorical MVAE participation groups. MMpW was considered statistically significant if the 95% CI for *B* did not include zero. There were missing MMpW values for $n = 338$ participants, so sensitivity analyses were completed in the subset of the study sample ($n = 1,002$). All study analyses were performed with SPSS v29.0 (Armonk, NY).

## Results

Participants in this study predominantly identified as non-Hispanic white (73.4%), male (87.0%), and with the majority having reported a lifetime history of at least 1 mTBI (80.7%; $M \pm SD = 2.2 \pm 2.0$; median [IQR] = 2 [1–3]; total range = 0 to 15). Approximately two-thirds of participants (66.0%) reported engagement in at least some weekly aerobic exercise as their primary physical activity, with about one-third (34.2%) meeting recommended MVAE recommendations (Table 1).

### Lifetime mTBI history

While accounting for age, sex, education, and PTSD symptoms, there were no statistically significant differences for any of the brain volume outcome measures for participants in the *1-2* or *3 +* lifetime mTBI history groups when compared to those with 0 lifetime mTBIs (Table 2; Fig 1).

### Participation in moderate-to-vigorous aerobic exercise

Participants reporting no current MVAE participation (*Inactive*) had significantly larger VBR values than those in the *Insufficiently Active* group while controlling for age, sex,

**Table 1. Participant characteristics.** Demographic, mild traumatic brain injury (mTBI) history, and moderate-to-vigorous aerobic exercise (MVAE) participation characteristics among the sample of participants with complete mTBI history, MVAE, and MRI data. Data are presented for the full sample and for each separate mTBI history group.

| | | Lifetime mTBI group | | |
|---|---|---|---|---|
| | Full sample n = 1,340 | 0 n = 259 | 1-2 n = 632 | 3 + n = 449 |
| Age in years, mean (sd) | 41.3 (10.3) | 41.7 (11.1) | 40.8 (10.3) | 41.8 (9.7) |
| Female sex, n (%) | 174 (13.0) | 57 (22.0) | 80 (12.7) | 37 (8.2) |
| Racial and ethnic identity, n (%) | | | | |
| American Indian or Native Alaskan | 15 (1.1) | 1 (0.4) | 7 (1.1) | 7 (1.6) |
| Asian | 15 (1.1) | 4 (1.5) | 8 (1.3) | 3 (0.7) |
| Black or African American | 232 (17.3) | 57 (22.0) | 131 (20.7) | 44 (9.8) |
| Pacific Islander | 10 (0.7) | 2 (0.8) | 4 (0.6) | 4 (0.9) |
| White | 984 (73.4) | 182 (70.3) | 443 (70.1) | 359 (80.0) |
| Did not specify racial identity[a] | 84 (6.3) | 13 (5.0) | 39 (6.2) | 32 (7.1) |
| Hispanic or Latino ethnicity[b] | 225 (16.8) | 48 (18.5) | 109 (17.2) | 68 (15.1) |
| Highest level of education completed, n (%) | | | | |
| High school diploma or less | 168 (12.5) | 37 (14.3) | 85 (13.4) | 46 (10.2) |
| Some college or technical school | 533 (39.8) | 96 (37.1) | 247 (39.1) | 190 (42.3) |
| College graduate or more | 639 (47.7) | 126 (48.6) | 300 (47.5) | 213 (47.4) |
| Posttraumatic stress screening status, n (%) | | | | |
| Positive (PCL-5[c] total > 33) | 451 (33.7) | 52 (20.1) | 220 (34.8) | 179 (39.9) |
| Meets MVAE guidelines, n (%) | 459 (34.2) | 84 (32.4) | 216 (34.2) | 159 (35.4) |
| MVAE categories, n (%) | | | | |
| Not aerobically active (Inactive; no aerobic activities) | 455 (34.0) | 82 (31.7) | 222 (35.1) | 151 (33.6) |
| Insufficiently active (aerobic activities, but didn't reach recommendation threshold) | 426 (31.8) | 93 (35.9) | 194 (30.7) | 139 (31.0) |
| Active (met recommended MVAE guidelines) | 236 (17.6) | 47 (18.1) | 112 (17.7) | 77 (17.1) |
| Highly active (at least twice the volume of MVAE as recommended) | 223 (16.6) | 37 (14.3) | 104 (16.5) | 82 (18.3) |
| MET-minutes per week (MMpW), median (IQR)[d] | 682 (360, 1102.5) | 630 (315, 1027.5) | 684 (360, 1242) | 720 (408, 1170) |
| Hazardous alcohol consumption (AUDIT-C), n (%)[e] | 455 (34.0) | 74 (28.6) | 217 (34.3) | 164 (36.5) |

[a]Unspecified racial identity: participants responded as "other", "don't know", or chose not to respond this item.

[b]Eighteen participants refused to respond or responded "don't know/not sure" to the question about Hispanic or Latino ethnicity.

[c]PCL-5 = Posttraumatic Stress Syndrome (PTSD) Checklist for the Diagnostic and Statistical Manual, fifth edition.

[d]MMpW was missing for a total of n = 350 participants: n = 61 (23.5%) in the no mTBI group; n = 179 (28.3%) in the 1-2 mTBI group; and n = 98 (21.8%) in the 3 + mTBI group.

[e]Alcohol consumption information was not provided by 5 (0.4%) of the final sample.

education, and PTSD symptoms (Table 2; Fig 1). There were no other statistically significant differences for VBR or any of the other brain volumetric outcomes between MVAE participation groups.

The overall interaction term between MVAE participation grouping and lifetime mTBI groups was statistically significant (Wald $X^2_{(6)}$ = 13.292; $p$ = 0.039) in the model for VBR. More specifically, in pairwise comparisons of marginal means between mTBI history and MVAE subgroups while accounting for age, sex, education, and PTSD symptoms, those in the *0 mTBI* group who were *Highly Active* had larger VBR values compared to those who were *Inactive* ($p$ = 0.035) and *Insufficiently Active* ($p$ = 0.046). Additionally, *Highly Active* participants in the *3 + mTBI* group had smaller VBR values than *Highly Active* participants in the *0 mTBI* ($p$ = 0.005) and *1-2 mTBI* ($p$ = 0.025) groups (Fig 1). There were no significant interaction effects observed for the remaining volumetric outcomes.

**Table 2. Main Effects General Linear Model Results for Primary Hypothesis Tests.** Values are unstandardized betas and 95% confidence intervals. Lifetime mTBI history groups were contrasted with the referent 0 lifetime mTBI history group. Moderate-to-vigorous aerobic exercise participation groups were contrasted with the no aerobic activity ('Inactive') group. All models included age, sex, education attainment, and PTSD symptoms as covariates.

| Volumetric outcome | Lifetime mTBI history | | Moderate-to-vigorous aerobic exercise participation | | |
| --- | --- | --- | --- | --- | --- |
| | 1-2 | 3 + | Insufficiently active | Active | Highly active |
| Ventricle-to-brain ratio | -0.013 (-0.149, 0.123) | -0.119 (-0.264, 0.027) | -0.139 (-0.263, -0.015)ᵃ | -0.033 (-0.181, 0.115) | -0.034 (-0.185, 0.117) |
| Frontal lobe volume | -0.110 (-0.244, 0.025) | -0.043 (-0.187, 0.101) | -0.007 (-0.130, 0.115) | -0.029 (-0.175, 0.117) | 0.106 (-0.043, 0.256) |
| Parietal lobe volume | -0.068 (-0.204, 0.068) | 0.003 (-0.143, 0.148) | -0.092 (-0.216, 0.031) | -0.031 (-0.179, 0.116) | 0.077 (-0.074, 0.228) |
| Temporal lobe volume | -0.026 (-0.157, 0.105) | -0.001 (-0.141, 0.138) | -0.042 (-0.161, 0.077) | -0.007 (-0.149, 0.135) | -0.003 (-0.148, 0.142) |
| Occipital lobe volume | 0.033 (-0.107, 0.173) | 0.080 (-0.070, 0.230) | -0.110 (-0.238, 0.017) | -0.067 (-0.219, 0.085) | -0.028 (-0.183, 0.128) |
| Cingulate cortex volume | -0.124 (-0.264, 0.017) | -0.113 (-0.263, 0.037) | -0.111 (-0.238, 0.017) | -0.016 (-0.168, 0.136) | 0.075 (-0.081, 0.231) |
| Hippocampus volume | -0.103 (-0.238, 0.031) | 0.006 (-0.138, 0.150) | 0.030 (-0.093, 0.152) | -0.030 (-0.176, 0.116) | -0.035 (-0.184, 0.115) |
| Amygdala volume | -0.050 (-0.191, 0.092) | -0.016 (-0.167, 0.135) | -0.022 (-0.151, 0.106) | 0.017 (-0.170, 0.137) | -0.002 (-0.159, 0.155) |
| Thalamus volume | -0.029 (-0.161, 0.104) | -0.046 (-0.187, 0.096) | 0.000 (-0.120, 0.121) | -0.085 (-0.229, 0.058) | -0.047 (-0.194, 1.000) |

ᵃStatistically significant as 95% CI does not include zero.

In the planned sensitivity analyses, more MMpW of aerobic exercise was a statistically significant predictor only for smaller thalamus volumes (*B* [95%CI] = -0.058 [-0.114, -0.002]; *p* = 0.044). Results for all other outcomes were not statistically significant (*p*-values > 0.16).

## Discussion

In this secondary analysis of a large, well-characterized sample of combat-exposed military SMVs, we observed no statistically significant associations between lifetime mTBI history and brain volumetrics. This lack of finding long-term detrimental effects of sustaining one or more mTBI on surrogate measures of brain reserve should be encouraging for SMVs who have sustained one or more mTBI. We also did not observe a consistent or clinically relevant pattern of association between engagement in MVAE activities and brain gray matter volumes. As such, none of our hypotheses were supported by our findings in this cross-sectional study. Future investigations will need to account for other important factors related to these predictors of brain health including, but not limited to, longitudinal changes in MVAE participation, exposure to repetitive head trauma that doesn't necessarily result in overt clinical signs and symptoms (e.g., low-level blast exposure), and measurement of brain health trajectories over time.

### Lifetime mTBI history

Our observed lack of statistical association between mTBI history and measures of brain gray matter volumes are in line with recent work from an overlapping sample of the LIMBIC-CENC PLS showing no significant cross-sectional associations between mTBI history and objectively measured neurocognitive test performance across multiple domains [14]. Other studies have failed to detect an association between mTBI history and objective cognitive decline in civilian samples including participants aged 50 years and older, when cognitive changes may be expected in relation to normal aging [15,52]. Taken together, these findings suggest that exposure to one or more mTBI does not typically result in long-term detriments to brain health based on brain volumetrics or cognitive test performance. However, it is important to also acknowledge clinically-pertinent nuances when interpreting these findings. For example, mTBI mechanism may be a salient factor in this population, as recent studies have noted significant associations of blast-related mTBI exposure with neurological and headache symptoms [53,54]. Additionally, there may be a subset of SMVs who are more likely

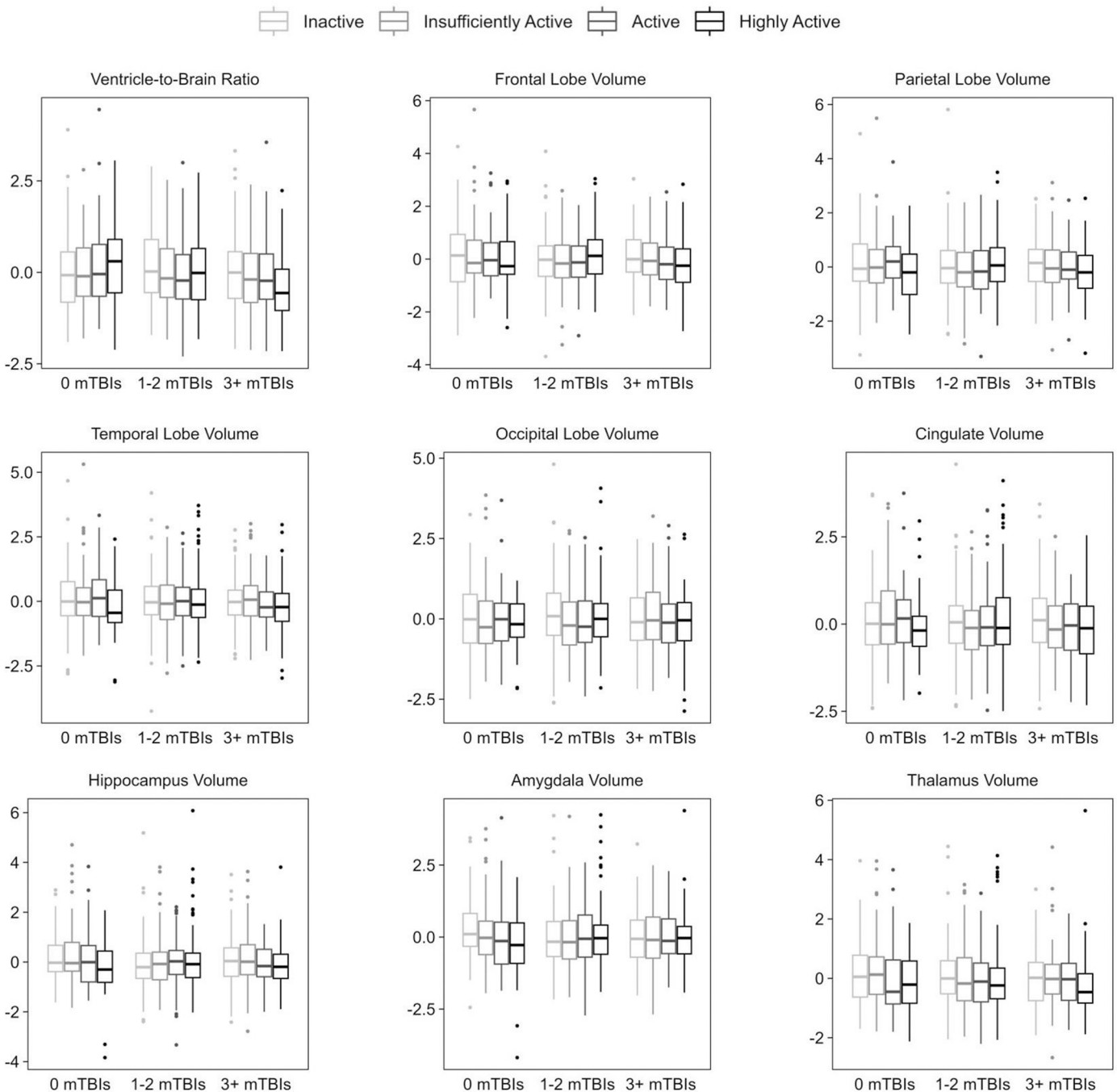

**Fig 1. Distribution of Normalized Brain Structure Volumes by mTBI History and participation in MVAE.** Y-axes represent z-scores for normalized brain volume measures. X-axes are demarcated by lifetime mTBI history group. Each MVAE participation category is shown within each mTBI history group, from left to right: *Inactive; Insufficiently Active, Active, Highly Active* (Legend). Boxplots include median value (middle horizontal line), 25%ile (lower horizontal line), 75%ile (upper horizontal line), whiskers (vertical lines; 1.5 times the interquartile range), and outliers (dots; beyond 1.5 times the interquartile range).

to be negatively impacted by mTBI history, but we did not explicitly examine this in our study.

Future consideration should be given to the potential for a dose-response relationship between mTBI exposure and brain health beyond the categorical operationalization used in

the present study (*0 mTBI*, *1-2 mTBIs*, *3+ mTBIs*). In our sample, the total number of lifetime mTBIs among those in the *3+ mTBI* group ranged from 3-15, and this may have affected the ability to identify a signal for dose-response in our study. As an example from another population with exposure to repetitive head injuries, studies with former professional American football players have reported statistical associations between multiple mTBI injuries and several clinical outcomes among those with 10 or more self-reported injuries, but not necessarily all participants with three or more lifetime mTBIs [11,55,56]. These observations in former professional football players suggest that there is still much to be learned about cumulative exposure to mTBI, the measurement (e.g., self-report versus structured interview) and operationalization of mTBI as a predictor variable, and long-term implications for brain health [56]. Further research is also needed to examine more specifically mTBIs with blast-related mechanism and to expand upon cross-sectional studies to understand trajectories of brain health over time and how these may be associated with lifetime exposure to mTBI.

## Participation in moderate-to-vigorous aerobic exercise

Similar to the null findings observed for mTBI history, participation in MVAE was observed to have no discernable pattern of association with brain gray matter volumes. In models for VBR, we observed that SMVs in the *Insufficiently Active* MVAE group had better outcome scores (smaller VBR values) compared to those in the *Inactive* group, but this effect was not observed for the *Active* or *Highly Active* MVAE groups, suggesting a lack of dose-response to engagement in weekly MVAE. Conversely, the interaction terms observed for MVAE and lifetime mTBI groups suggested that *Highly Active* SMVs and SMVs with fewer lifetime mTBIs had more better VBR outcome scores (smaller VBR values) when compared to less active SMVs and SMVs with more lifetime mTBIs. These observations are antithetical to our hypotheses regarding the anticipated effects of lifetime mTBI history and participation in MVAE. With no other significant interaction effects across all measures, it is possible that these findings are related to type I errors, where significant findings were observed that may not actually represent the true nature of this relationship. It will be important to study the relationship between MVAE participation and mTBI history among SMV's in future research studies, as there may be subsets of individuals in this population that have real or perceived barriers to MVAE engagement stemming from ongoing symptoms (e.g., headache), disability (e.g., physical impairment), or comorbidities (e.g., cardiovascular disease, mood disorder) that weren't directly assessed in the present study.

Although our results regarding MVAE participation are similar to those recently reported for objectively measured neurocognitive test performance in an overlapping LIMBIC-CENC PLS cohort sample [57]. The current cross-sectional study findings in SMVs do not align with reports of preserved brain structure and cognitive functioning in the general population for youth and adults participating in aerobic exercise [22–25,27,29]. One source for these disparate findings may have been our method of measuring MVAE as a predictor variable. Participants reported primary physical activities on the BRFSS, most of which were able to be cross-walked to identify MET values and generate MVAE categories based on these reports; however, secondary aerobic activities, non-aerobic exercise, and non-leisure-time activities were not well characterized, resulting in an incomplete picture of participants' overall engagement in physical activities. Of note, participation in resistance training (e.g., lifting weights) has been associated with benefits to brain structure and function [58], but this form of physical activity was not included in the present study. Future studies with more objective measures of daily physical activities and exercise (e.g., via wearable activity monitors) will likely provide a better understanding of the physiological adaptations to exercise participation. Additionally, measurement of physical activity over a longer period of time may help to inform clinical

interventions with exercise given that measures of cardiorespiratory fitness [29], not just participation in exercise, may be the primary exercise-related factor in preserving brain health as people age. Finally, accounting for lifetime exercise participation (sometimes referred to as *motor reserve*) [59] alongside more recent engagement in exercise might aid in understanding the short- and long-term benefits of exercise on brain health in this population.

## Future clinical and research considerations

**Participants.** All participants in the LIMBIC-CENC PLS were eligible for inclusion in the present analyses; however, it's possible that there are subsets of this sample (and the underlying SMV populations) who would be more affected by mTBI history and/or MVAE participation. For example, individuals who are chronically symptomatic after mTBI could be less likely to engage in regular weekly MVAE, but may also stand to benefit the most from the subjective and physiological benefits of exercise. If true, this association may not have been detectable by including all eligible participants, regardless of their symptomatic status. In an overlapping cross-sectional study investigating associations between MVAE participation and cognitive outcomes, the authors reported potentially meaningful associations of higher MVAE participation with self-reported global health status and life satisfaction outcomes [57]. That said, the direction of associations between MVAE participation and health-related factors (i.e., temporality or causation) cannot be determined from our cross-sectional data. There is still much to be learned about barriers and facilitators to MVAE participation in SMVs, especially after leaving the structure of military service and accompanying exercise requirements. Given the exercise requirements during active service, it is also possible that study participants who sustained mTBI(s) during active periods benefitted from MVAE during recovery as has been noted in in prior studies. We are presently unable to account for acute management strategies employed for injuries sustained by our study participants, in many cases, years prior to enrolling in this study. Future research should also focus on characterizing engagement in MVAE and other health behaviors as factors that can be targeted to potentially improve brain health in both acute and long-term care settings.

**Study Design.** Cross-sectional studies such as ours can be helpful for identifying patterns of association, but they are not well suited to account for changes in exposures, behaviors, or brain health outcomes over time. Results from our study suggest that assessment of brain gray matter outcomes are not useful for tracking treatment response to MVAE in the short term; however, longitudinal study of aerobic and other exercise participation over months or years may glean more meaningful information regarding measures of brain reserve. For example, MVAE interventions have been reported to improve brain structure and function over the course of many months, noting the benefits of improved aerobic fitness beyond engagement in exercise alone [27,29]. At the same time, accounting for historical, lifetime participation in exercise might help us understand exercise benefits and prescription in clinical populations [59]. Use of wearable sensor technologies, such as a wrist-worn activity monitors, will help better capture and characterize physical activity (and inactivity) over the course of a longitudinal study, and may garner better information regarding brain health outcomes.

**Brain Health Outcomes.** While we did not find an association of exercise on brain morphological measurements in the current study, we cannot discount this surrogate measurement of brain reserve in future longitudinal studies of physical activity in this population. It is plausible that there were more functional/neurophysiological changes in response to exercise (e.g., adaptive neurometabolic activity) among this study sample – outcomes that were not included in the current investigation. Based on the limitations of the current study and the potential for both structural and functional changes with exercise,

we recommend that scientists continue to use multimodal measurements of both brain morphometry and neurophysiological outcomes, as well as subjectively reported functioning, in the evaluation of exercise as a behavioral promoter of brain health for SMVs.

## Limitations

The most notable limitation of this exploratory study was the truncated and subjective measurement of MVAE. As presented above, self-reported primary leisure time activities acted as a surrogate measure for overall engagement in aerobic exercise but were limited by inconsistent reporting of other physical activities and lack of objective measurement. More detailed subjective reports and/or objective data captured with wearable physical activity monitors will help to better describe the role of physical activity in the promotion of brain health in this population. Additionally, the measurement of mTBI history in this study was rigorous and robust, but we did not objectively measure injuries at the time of occurrence, and we did not account for exposure to other low-level blast-related or other blunt-force repetitive head traumas that could potentially affect brain physiology or morphometry without overt clinical signs or symptoms. Further, as a cross-sectional study, we cannot infer whether mTBI history or sustained participation in MVAE affect trajectories of brain health over time. These are important factors that are in need of further investigation in future studies.

## Conclusions

History of one or more lifetime mTBIs was not associated with long-term measures of brain gray matter volumes, suggesting that declines in structural brain health are not inevitable for service members who sustain these injuries. There was also no clear pattern of positive association between self-reported MVAE participation and brain gray matter volumes. While we did not identify an overall benefit from regular engagement in MVAE, it remains a consensus recommendation that exercise participation is an important and practical means to promote other facets of health (e.g., cardiovascular function, mood-related symptoms, satisfaction with life, etc.) in SMVs and the broader civilian population. Our study adds to the nascent literature that serves as a starting point for further investigations into the potential roles of mTBI exposure and engagement in aerobic physical activity as modifiers of brain health.

## Supporting information

**S1 Supplemental Materials. Contains "Supplemental Methods" which show comparisons of demographic, MVAE, and mTBI characteristics between those with MRI data available for this study (n = 1,349) vs. those without (n = 902).** Also contains "Supplemental Table 1. Gr**ay matter variables from FreeSurfer v7.1.1 processing**" detailing all individual region of interest labels and formulas for aggregate regionns of interest based on those labels. (DOCX)

ACKNOWLEDGEMENTSWe are grateful to the myriad staff, faculty, advocates, service members, Veterans, and other team members and stakeholders who make the LIMBIC-CENC PLS possible across all our research sites and support investigations such as this.

This investigation was supported by the University of Utah Study Design and Biostatistics Center, with funding in part from the National Center for Research Resources and the National Center for Advancing Translational Sciences, National Institutes of Health, through Grant UL1TR002538 (formerly 5UL1TR001067-05, 8UL1TR000105 and UL1RR025764).

## Author contributions

**Conceptualization:** Samuel R. Walton, John J. Fraser, Hannah M. Lindsey, Naomi J. Goodrich-Hunsaker, Elisabeth A. Wilde, Randel L. Swanson, William C. Walker.

**Data curation:** Samuel R. Walton, Hannah M. Lindsey, Naomi J. Goodrich-Hunsaker, Emily L. Dennis, Elisabeth A. Wilde, David F. Tate, William C. Walker.

**Formal analysis:** Samuel R. Walton, Hannah M. Lindsey, Naomi J. Goodrich-Hunsaker.

**Funding acquisition:** Elisabeth A. Wilde, David X. Cifu, William C. Walker.

**Investigation:** Elisabeth A. Wilde, David X. Cifu, William C. Walker.

**Methodology:** Samuel R. Walton, John J. Fraser, Jessie R. Oldham, Mark L. Ettenhofer, Patrick Armistead-Jehle, Hannah M. Lindsey, Naomi J. Goodrich-Hunsaker, Emily L. Dennis, Elisabeth A. Wilde, Sarah M. Jurick, Michael J. Hall, Randel L. Swanson, Andrew J. MacGregor, David F. Tate, David X. Cifu, William C. Walker.

**Project administration:** Elisabeth A. Wilde, David F. Tate, David X. Cifu, William C. Walker.

**Resources:** Hannah M. Lindsey, Naomi J. Goodrich-Hunsaker, Emily L. Dennis, David X. Cifu, William C. Walker.

**Supervision:** David X. Cifu, William C. Walker.

**Visualization:** Samuel R. Walton.

**Writing – original draft:** Samuel R. Walton, John J. Fraser, Jessie R. Oldham, Mark L. Ettenhofer, Patrick Armistead-Jehle, Hannah M. Lindsey, Naomi J. Goodrich-Hunsaker, Emily L. Dennis, Sarah M. Jurick, Andrew J. MacGregor.

**Writing – review & editing:** Samuel R. Walton, John J. Fraser, Jessie R. Oldham, Mark L. Ettenhofer, Patrick Armistead-Jehle, Hannah M. Lindsey, Naomi J. Goodrich-Hunsaker, Emily L. Dennis, Elisabeth A. Wilde, Sarah M. Jurick, Michael J. Hall, Randel L. Swanson, Andrew J. MacGregor, David F. Tate, David X. Cifu, William C. Walker.

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
