## [Decision Letter · Decision Letter 0]

12 Feb 2025

Aerobic Exercise and Brain Structure among Military Service Members and Veterans with varying histories of Mild Traumatic Brain Injury: A LIMBIC-CENC Exploratory Investigation

PONE-D-24-08461

Dear Dr. Walton,

We’re pleased to inform you that your manuscript has been judged scientifically suitable for publication and will be formally accepted for publication once it meets all outstanding technical requirements.

Kind regards,

Gary B. Wilkerson, EdD

Academic Editor

PLOS ONE

1.  Thank you for stating the following financial disclosure:

“MLE, SMJ, JJF, AJM, PAJ are military service member employees of the United States government. This work was prepared as part of their official duties. Title 17 U.S.C. 105 provides that copyright protection under this title is not available for any work of the United States Government. This work was supported by the Assistant Secretary of Defense for Health Affairs endorsed by the Department of Defense, through the Psychological Health/Traumatic Brain Injury Research Program Long-Term Impact of Military-Relevant Brain Injury Consortium (LIMBIC) Award/W81XWH-18-PH/TBIRP-LIMBIC under Awards No. W81XWH1920067 and W81XWH-13-2-0095, and by the U.S. Department of Veterans Affairs Awards No. I01 CX002097, I01 CX002096, I01 HX003155, I01 RX003444, I01 RX003443, I01 RX003442, I01 CX001135, I01 CX001246, I01 RX001774, I01 RX 001135, I01 RX 002076, I01 RX 001880, I01 RX 002172, I01 RX 002173, I01 RX 002171, I01 RX 002174, and I01 RX 002170. The U.S. Army Medical Research Acquisition Activity, 839 Chandler Street, Fort Detrick MD 21702-5014 is the awarding and administering acquisition office. The study protocol was approved by the Virginia Commonwealth University Institutional Review Board in compliance with all applicable federal regulations governing the protection of human subjects, protocol number HM20002321. The views, opinions, interpretations, conclusions and recommendations expressed are those of the authors and do not necessarily reflect the official policy of the Department of the Navy, the Department of the Army, Department of Defense, Department of Veterans Affairs or the U.S. Government.”

Please respond by return e-mail so that we can amend your financial disclosure and competing interests on your behalf.

“I have read the journal's policy and the authors of this manuscript have the following competing interests: SRW declares separate funding support from the Departments of Defense and Veterans affairs for projects related to the LIMBIC-CENC study; honorarium and conference travel support from the National Athletic Trainers’ Association and the University of California – Los Angeles; and serves in unpaid leadership roles with the World Federation of Athletic Training and Therapy (WFATT) and the Concussion in Sport Group (CISG). JJF reports grants from Congressionally Directed Medical Research Programs and the Office of Naval Research, outside of the submitted work; and has a patent pending for an Adaptive and Variable Stiffness Ankle Brace, U.S. Provisional Patent Application No. 63254,474. JRO receives support from the Thrasher Research Fund and National Institute of Arthritis and Musculoskeletal Skin Diseases (8K12AR084233-03). EAW reports grants from Congressionally Directed Medical Research Programs, US Department of Veterans Affairs, and National Institutes of Health, outside of the submitted work. RLS was supported, in part, by the US Department of Veterans Affairs Rehabilitation Research and Development Service under award number IK2 RX003651, for an independent project related to the LIMBIC-CENC study. DFT, DXC, and WCW report separate funding support from the DoD & VA for projects related to the LIMBIC-CENC study. The remaining authors have no disclosures of potential conflicts of interest to report.”

Please respond by return email with your amended Competing Interests Statement and we will change the online submission form on your behalf.

Additional Editor Comments (optional):

After having thoroughly reviewed the manuscript content multiple times, I have not found any flaws in the data analysis or interpretation of the findings. Neither did I find any grammatical errors or technical inaccuracies that need revision. In fact, I commend the authors for having provided an extremely well-structured report that does not require responses to specific comments or questions. The manner in which the information has been formatted is appopriate and the written presentation provides a logical flow of ideas. Despite the lack of any hypothesized differences that were statistically significant, the null findings have value for interpretation of the larger body of evidence on the topic. My only comment relates to the exclusive focus of the Conclusions on gray matter volumes as indicators of structural brain health. Although you did not assess the microstructural integrity of white matter tracts, the possiblility that abnormalities in structural connectivity might co-exist with normal gray matter volumes should be acknowledged. Overall, the work represents a worthy contribution to the literature that may reduce worry that multiple mTBIs will inevitably result in deterioration of brain health.

Reviewers' comments:

Reviewer's Responses to Questions

**Comments to the Author**

1. Is the manuscript technically sound, and do the data support the conclusions?

Reviewer #1: No

2. Has the statistical analysis been performed appropriately and rigorously? 

Reviewer #1: Yes

3. Have the authors made all data underlying the findings in their manuscript fully available?

Reviewer #1: Yes

4. Is the manuscript presented in an intelligible fashion and written in standard English?

Reviewer #1: Yes

5. Review Comments to the Author

Reviewer #1: This is a well-conducted study that explores the associations of MVAE participation and lifetime mTBI history with measures of brain gray matter volumes among SMVs. The study uses a large sample and rigorous methods. While the findings are largely null, they are still important and contribute to the literature on mTBI and exercise.

6. PLOS authors have the option to publish the peer review history of their article (what does this mean? ). If published, this will include your full peer review and any attached files.

**Do you want your identity to be public for this peer review?** For information about this choice, including consent withdrawal, please see our Privacy Policy .

Reviewer #1: **Yes: ** Joseph D. Walker

---

## [Editor Report · Acceptance letter]

PONE-D-24-08461

PLOS ONE

Dear Dr. Walton,

I'm pleased to inform you that your manuscript has been deemed suitable for publication in PLOS ONE. Congratulations! Your manuscript is now being handed over to our production team.

Kind regards,

on behalf of

Prof. Gary B. Wilkerson

Academic Editor

PLOS ONE